# Phylogenomic and Evolutionary Analyses Reveal Diversifications of SET-Domain Proteins in Fungi

**DOI:** 10.3390/jof8111159

**Published:** 2022-11-02

**Authors:** Guoqing Ding, Liqiu Shang, Wenliang Zhou, Siyi Lu, Zong Zhou, Xinyi Huang, Juan Li

**Affiliations:** State Key Laboratory for Conservation and Utilization of Bio-Resources in Yunnan, Yunnan University, Kunming 650091, China

**Keywords:** histone methyltransferases, Ascomycota, gene duplication and loss, methylation, epigenetic modification, collinearity relationships

## Abstract

In recent years, many publications have established histone lysine methylation as a central epigenetic modification in the regulation of chromatin and transcription. The histone lysine methyltransferases contain a conserved SET domain and are widely distributed in various organisms. However, a comprehensive study on the origin and diversification of the SET-domain-containing genes in fungi has not been conducted. In this study, a total of 3816 SET-domain-containing genes, which were identified and characterized using HmmSearch from 229 whole genomes sequenced fungal species, were used to ascertain their evolution and diversification in fungi. Using the CLANS program, all the SET-domain-containing genes were grouped into three main clusters, and each cluster contains several groups. Domain organization analysis showed that genes belonging to the same group have similar sequence structures. In contrast, different groups process domain organizations or locations differently, suggesting the SET-domain-containing genes belonging to different groups may have obtained distinctive regulatory mechanisms during their evolution. These genes that conduct the histone methylations (such as H3K4me, H3K9me, H3K27me, H4K20me, H3K36me) are mainly grouped into Cluster 1 while the other genes grouped into Clusters 2 and 3 are still functionally undetermined. Our results also showed that numerous gene duplication and loss events have happened during the evolution of those fungal SET-domain-containing proteins. Our results provide novel insights into the roles of SET-domain genes in fungal evolution and pave a fundamental path to further understanding the epigenetic basis of gene regulation in fungi.

## 1. Introduction

Histones are subject to numerous post-translational modifications, including acetylation, methylation, phosphorylation, ubiquitylation, sumoylation, and ADP-ribosylation [1]. In recent years, histone modifications, especially histone lysine methylation, have been demonstrated as a significant epigenetic modification in the regulation of gene expression and chromatin organization [2,3,4]. Histone lysine methylation is usually catalyzed by the histone lysine methyltransferases (HMTs) with an evolutionarily conserved motif named SET domain, with the exception of Dot1, which mediates the H3K79 methylation [5,6]. The SET domain stands for “Suppressor of variegation, Enhancer of zeste and Trithorax”, which are the three genes initially derived from *Drosophila melanogaster* [7,8]. At present, SET-domain-containing genes have been found in various organisms ranging from Prokaryotes to Eukaryotes [9]. The SET domain is mainly comprised of approximately 130-150 amino acid residues, which are capable of catalyzing mono-, di-, or tri-methylation of their lysine substrates, and lysine residues in different histones that can be methylated by the SET domain including K4 (K is the abbreviation for lysine), K9, K27, K36, and K79 in histone H3; K20 in histone H4; K59 in the globular domain of histone H4; and K26 in histone H1B [10]. The methylation of lysine residues in histone protein can serve as an epigenetic marker to regulate gene expressions at the post-translational level. Diverse biological mechanisms such as transcriptional activation and repression, enhancer function, mRNA splicing, and DNA replication have all been associated with SET-domain-containing genes [3,11,12].

In fungi, the modification of histone N-terminal tails is overall conserved. In general, the histone modification landscape usually generates as transcriptionally active or transcriptionally repressive. The cross-talk between various modification machineries formed interconnected positive and negative feedback loops that control gene regulation [13]. It has been frequently asserted that H3K9me is associated with the silent regions of both euchromatin and heterochromatin, and responsible for transcriptional repression [14,15,16], while H3K4me is related exclusively to actively transcribed genes [17]. In addition, H3K36me, which is catalyzed by SET2, is also associated with gene activity [18]. Moreover, trimethylation of histone H3 lysine 27 (H3K27me3) has been recognized as important for differentiation and development in fungi [19,20]. Aside from its relatively well-understood role in transcriptional repression, accumulating evidence suggests that H3K27 methylation has an important role in controlling the balance between maintenance and generation of novelty in fungal genomes [21]. Those histone lysine modifications have been extensively investigated to play vital roles in fungal development and various biological processes. For instance, in *Neurospora crassa*, a histone H3K9 methyltransferase Dim-5 was reported to be required for normal growth and full fertility of *N. crassa* [22]. NcSet2, the sole lysine histone methyltransferase responsible for the methylation of H3K36, is essential for normal growth and development in *N. crassa* [23]. ScSet4 has been characterized as a regulated chromatin factor required for controlling gene expression in response to stress in *Saccharomyces cerevisiae* [24]. MoSet1, a histone H3K4 methyltransferase in the phytopathogenic fungus *Magnaporthe oryzae*, has been confirmed to regulate global gene expression during infection-related morphogenesis [25]. FvSet1 was also found to play an important role in vegetative growth, virulence, and environmental stress responses in the plant pathogen *Fusarium verticillioides* [26]. All these studies provided strong evidence that SET-domain-containing methyltransferases play crucial roles in regulating a variety of biological processes in fungi, including growth, differentiation, development, metabolic processes, and environmental adaptation [27,28]. 

Although the importance of SET-domain containing methyltransferases in fungi has been confirmed, the SET-domain genes have only been identified in a limited number of representative fungal species, primarily in model filamentous fungi such as *N. crassa* and *M. oryzae* and in model yeasts *S. cereviseae* and *Schizosaccharomyces pombe* [4,29]. Similarly, studies related to the evolution of SET-domain genes in fungi have been limited to a few fungal species [30]. Thus, the pattern of conservation and diversification of SET genes among fungal species remains largely unknown. Therefore, in this study, we conducted a large-scale phylogenomic survey of SET-domain genes from 229 fungal species, which cover all the phyla within the fungal kingdom. We aimed to identify the evolutionary dynamics and functional divergences of SET-domain genes among diverse fungal species. Our results provide a comprehensive understanding of the evolution of SET-domain genes in fungi.

## 2. Materials and Methods

### 2.1. Identification and Homologous Searches of SET-Domain Containing Genes in Fungi

To find all the putative SET-domain containing sequences from fungi, the HMM profile SET domain (PF00856; http://pfam.xfam.org/family/PF00856, accessed on 1 August 2022) was downloaded and used as a query with E values <10^−5^ in the program and HMMSEARCH from the HMMER package (http://hmmer.wustl.edu/, accessed on 4 August 2022) to search the homologous proteins in 229 fungal genome sequences. All 229 fungal genomes were available from the Mycocosm of Joint Genome Institute (https://mycocosm.jgi.doe.gov/mycocosm/home, accessed on 27 July 2022), Fungal Genome Research (http://fungalgenomes.org/, accessed on 27 July 2022), and the NCBI database. This is the first analysis of SET-domain-containing genes covering such a range of fungal species with different lifestyles.

### 2.2. Distribution, Structure Identification of SET-Domain Containing Genes in Fungi

A Java-based software program, CLANS, was used to elucidate the relationships between and within subfamilies of the Hmmsearch results [31]. After the identifications of groups, the conserved functional domain structures of some representative sequences of each group were predicted using the InterProScan 5.0 (http://www.ebi.ac.uk/interpro/, accessed on 1 August 2022, Hinxton, UK) and MEME suite (https://meme-suite.org/meme/, accessed on 1 August 2022, Seattle, WA, USA) with default parameter settings. We combined both the two analysis results to figure out the domain structures of the SET-domain containing proteins in each cluster. 

### 2.3. Phylogenetic Analysis of SET-Domain Containing Genes in Fungi

Before phylogenetic analysis, MUSCLE v3.7 was used to generate protein alignment with default settings [32]. Then, the aligned sequences for each group were analyzed using two different tree construction methods: Neighbor-joining (NJ) analysis and Maximum likelihood (ML) analysis. For NJ analysis [33], the data were analyzed using MEGA X (Hachioji, Japan) [34] using bootstrap analysis (BS) with 1000 replicates. For ML analysis, the best-fit models of protein evolution for each subfamily were first estimated using the program ProtTest [35], then the recommended models and parameters were used for ML analysis with FastTree [36].

### 2.4. Analysis of Family Size over Evolutionary Time

In order to identify the expansions and contractions over evolutionary time about the five groups (Group 2, 4, 5, 6, and 12), which conduct the histone methylations of H3K27me, H3K4me, H3K9me, H3K36me, H4K20me, respectively, we chosen 20 representative fungal species to compare their family size for each group using the program Computational Analysis of gene Family Evolution v5 that use phylogenomics and family sizes to estimate the timing of gene family evolution (CAFE5; https://github.com/hahnlab/, accessed on 1 August 2022, CAFE5, Gothenburg, Sweden) [37,38]. Ultrametric phylogenies were generated using Orthofinder v2.3.11 (Oxford, UK) [39] and TimeTree was generated by the r8s [40].

### 2.5. Collinearity Analysis of SET-Domain Genes in Specific Fungal Species

To understand the collinearity relationships of the SET-domain gene family, five extensively studied Ascomycota fungal species, including three Eurotiomycetes fungi *Aspergillus fumigatus*, *Aspergillus niger*, and *Penicillium chrysogenum*, and one Sordariomycetes fungus *N. crassa* and one yeast species from Taphrinomycotina (*S. pombe*), with their genomes chromosomal-level assembled, were investigated using the TBtools (Guangdong, China) with MCScanX (Athens, GA, USA) [41,42]. In addition, using TBtools [41], we also visualized the chromosomal distribution of the SET-domain genes in the *A. fumigatus* genome.

## 3. Results and Discussions

### 3.1. Distribution of SET-Domain-Containing Genes in Fungi

We analyzed the SET-domain-containing genes in the fungi according to the flow chart shown in Figure 1. Firstly, we downloaded the HMM profile of the SET-domain from http://pfam.xfam.org/family/PF00856 and used this profile as a query model in the program HMMSEARCH. A total of 4172 SET-domain proteins were identified from 229 fungal species with different lifestyles with E values <10^−5^. To elucidate the relationships among these SET-domain sequences, a clustering analysis that relied on the sequence similarities of proteins was conducted using the CLANS program [31]. CLANS is a Java-based software program that visualizes pairwise sequence similarities in either two-dimensional or three-dimensional space [31]. In cases where a sequence showed low sequence similarities with other sequences, it would be set as a separate point in the CLANS program. Thus, those sequences that showed low sequence similarities with other sequences would be formed as the separated points in the CLANS program [31]. As seen in Appendix A, several sequences cannot group together with any other genes. For example, the *Jumonji C* (*JmjC*) genes identified from a few Pezizomycetes species, which have been identified to play roles in the modulation of histone methylation marks [43], showed low sequence similarities with each other and have been set as separated points in the CLANS program. Thus, those sequences, including JmjC-SET genes, are not considered in the following analysis because they cannot be well-grouped into any families. Finally, a total of 3816 SET-domain genes were used to conduct subsequent cluster analysis in this study, after removing the separated points in the CLANS program (Figure 2). 

The taxonomic information and the identified number of SET-domain-containing genes from each species and group are presented in Appendix A. Specifically, those 3816 SET-domain genes from 229 fungal species can mainly be classified into three distinct clusters based on amino acid sequence similarities (Cluster 1, 2, and 3, respectively), and 12, 6, and 9 groups can be observed in each of the three clusters (Figure 2). Remarkable variations in the number of SET-domain-containing genes were observed among different fungal species (Appendix A). A considerable variation in numbers was found within the Ascomycota, Basidiomycota, Chytridiomycota, Mucoromycota, Zoopagomycota, and Microsporidia fungi, ranging from 0 to 43 (Appendix A). The Basidiomycota fungus *Coprinopsis cinerea* contains 43 SET-domain sequences, whereas no specific SET-domain gene was identified from several Microsporidia fungi. The explanation for why those ancestral Microsporidia fungi did not contain any SET-domain genes may be associated with their genome size. As we can see in Appendix A, all the genome sizes of Microsporidia fungal species were less than 10Mb, suggesting that these species have only retained those essential functional genes to maintain their lives. In addition, scientists have found that the yeast *Cryptococcus neoformans* has lost the methylase gene between 50–150 mya, but the methylation in *C. neoformans* has persisted because an epigenome has been propagated for >50 million years through a process analogous to the Darwinian evolution of the genome [44,45]. Thus, although it is still hard to explain how the lysine methyltransferase activity could work in the fungi without the SET-domain gene, we supposed that there may be some uncovered mechanisms for those fungi to activate their lysine methyltransferase. Among the Pezizomycotina fungi of Ascomycota, the majority contain more than 10 SET-domain-encoding genes, and the Leotiomycetes fungus *Acephala macrosclerotiorum* contains 41 SET-domain genes, which is the largest number identified in Ascomycota fungi. For the yeast-like fungi of Ascomycota fungi, the gene number of most of the Saccharomycotina fungi is less than 10. In contrast, the total number of the SET-domain genes in each Taphrinomycotina fungi is larger than 10, suggesting that there may have been a constraint on the number of SET-domain genes after the split of Saccharomycotina and Taphrinomycotina fungi. These variations imply that gene duplication and loss may have played significant roles during the evolution of the fungal SET-domain-containing genes.

### 3.2. Conserved Motifs and Residues in SET Domain Proteins of Fungi

A protein sequence alignment of the SET domains from several representative histone lysine methyltransferases (HKMT) grouped according to their histone-lysine specificity showed that they contain conserved short conserved binding and catalytic motifs. According to the solved structures of the SET domain, we found that the SET-domain sequences contained the two most-conserved sequence motifs (RFINHXCXPN and ELXFDY, see Figure 3) binding to S-adenosylmethionine (AdoMet) and the target lysine of the SET domain. Moreover, the residues of the catalytic site; an intra-molecular interacting salt bridge; and a F/Y switch controlling whether the product is a mono-, di-, or tri-methylated histone were also identified. Previous studies supposed that the two conserved motifs binding to AdoMet were brought together by the “pesudoknot” fold to form an active site in a location immediately next to the pocket where the methyl donor binds to the peptide-binding cleft. Most notably, the substrate-binding clefts are connected through a deep channel that runs through the core of the SET domain and permits the transfer of the methyl group from AdoMet to the -amino group of the lysine substrate. 

### 3.3. Collinearity Analysis of SET-Domain Genes in Specific Fungal Species 

Subsequently, using a Toolkit for Biologists integrating various biological data-handling tools (TBtools) with the Multiple Collinearity Scan toolkit (MCScanX) method [41,42], we explored the collinearity relationships of the SET-domain genes among five chromosomal-level assembled Ascomycota species, including *A. fumigatus*, *Aspergillus niger*, *P. chrysogenum*, *N. crassa*, and *S. pombe* (Figure 4). Collinearity is a more specific form of synteny, which requires conserved gene order [42]. In our analysis, a total of 5895 gene pairs showed collinearity relationships between *A. fumigatus* and *A. niger*. In addition, 5642 gene pairs showed collinearity relationships between *A. fumigatus* and *P. chrysogenum*. Moreover, 13 SET-domain genes from *A. fumigatus* have collinearity relationships with *A. niger*, and *P. chrysogenum* (Figure 4). The great collinearity relationships among *A. fumigatus*, *A. niger*, and *P. chrysogenum* are consistent with their closed phylogenetic relationships. However, although 455 gene pairs showed collinearity relationships between *A. fumigatus* and *N. crass*, only one SET-domain gene pair was identified from these two species (Figure 4). Moreover, no SET-domain gene pairs were identified between *A. fumigatus* and *S. pombe*, which can be explained by their far phylogenetic relationships. Using *A. fumigatus* genome annotation information and TBtools [41], we also visualized the chromosomal distribution of the SET-domain-containing genes. Results from Appendix A indicate that the 13 SET-domain genes derived from *A. fumigatus* are unevenly distributed on the eight chromosomes, and the number of genes on each chromosome is irrelevant to chromosome size. Moreover, no segmental duplication events occurred among all the identified SET-domain genes in *A. fumigatus* (Appendix A). 

### 3.4. Characterization of Each Identified SET-Domain-Containing Gene in Fungi

To study the evolution of the SET-domain genes in fungi, we conducted a phylogenetic analysis based on the SET-domain genes identified from 20 representative fungal species according to their taxonomy classification. Phylogenetic trees were constructed based on the three distinct clusters of these SET-domain genes (Figure 5, Figure 6, and Figure 7). To show the relationships with SET-domain genes identified in previous studies and attempt to make the names of the SET genes less confusing, we used a nomenclature system according to previous research on the SET domain proteins in fungi. In addition to the families reported before, we also found some novel groups identified here.

#### 3.4.1. Cluster 1

Cluster 1 was comprised of 12 distinct groups. The genes in this cluster are known to catalyze di- and tri-methylation of H3K27, H3K4, H3K9, H3K36, and H4K20 (Group 2, 4, 5, 6, and 12, respectively) (Figure 5). Phylogenetic analysis of Cluster 1 genes identified from 20 representative fungal species showed that these 12 groups clustered into distinct clades with each other (Figure 5). Also, the functional domain structures of each group showed that the genes within the same group shared similar motif features while different group genes had different motif elements.

##### The SET3/4 Family (Group 1)

This group of proteins was commonly present in most of the fungal lineages except for Chytridiomycota (Figure 8). Structure analysis showed that proteins belonging to this group contain the conserved SET domain and a conserved PHD (plant homeodomain) finger domain at their N-terminal (Figure 5). Previously identified *ScSet3* (*Sc-NP_012430*), *ScSet4* (*Sc-NP_012954*) from *S. cerevisiae*, *NcSet4* (*Nc-XP_957466*) from *N. crassa*, and *SpSet3* (*Sp-NP_594837*) from *S. pombe* were clustered in this group (Table 1 and Figure 5). A previous study suggested that the ScSet3 and NcSet4 were components of a conserved histone deacetylase complex (HDAC), which can bind histone H3 dimethylation at lysine 4 (H3K4me2) to mediate deacetylation of histones in 5′ transcribed regions via their PHD finger [46]. In yeast, ScSet3 is involved in the meiosis-specific repression of sporulation genes [47] and is required for cellular protection against oxidative stress [24]. *C. albicans* Set3C is required for efficient biofilm production and drug resistance [48]. 

##### The Enhancer of Zeste (EZ) Family (Group 2)

This group’s proteins were also named the Enhancer of Zeste (Ez) family because they are components of the Polycomb PRC2 complexes, which play as a transcriptional suppressor for genes. The structures of the E(z)-type group genes include a CXC domain and a SET domain, whereas the *Nc-XP_965043* from *N. crassa* further processes another EZH2 domain (Figure 5 and Table 1). Earlier studies reported the absence of this group’s genes in the Saccharomycotina and the *S. pombe* genomes. However, in our analysis, we found this group’s genes from most of the Pezizomycotina fungi and identified several genes from Basidiomycota and the Chytridiomycetes and Microsporidia fungi. In contrast, no *E(z)*-type gene was found in Mucoromycota, Zoopagomycota, and the yeast-like fungi employed in our study (Appendix A and Figure 5). CAFE results from 20 fungal species showed that this group’s genes may have been lost before the split of Saccharomycotina and Taphrinomycotina fungi. Moreover, this group’s genes may have been lost by the ancestor of Basidiomycota but the Agaricomycetes fungi gained another gene copy during their evolution (Figure 9). Also, gene duplication has occurred in the fungus *Armillaria gallica*. Furthermore, we also found that several Eurotiomycetes fungi have lost the *E(z)*-type gene during their evolution (Figure 9). At present, the E(z) methylases have been reported to specifically mediate the H3K27me3 within Polycomb PRC2 complexes [9,49]. Thus, the Group 2 genes are predicted to account for the deposition of H3K27me3. 

##### The ASH1-Like Family (Group 3)

The genes in Group 3 were defined as an ASH1-like family because their two recognizable structural motifs (cysteine-rich peptide AWS, and SET domain) show similar motif structures to the *ASH1* genes of *Drosophila melanogaster* but lack AThooks and other motifs found in the *Drosophila ASH1* protein (Figure 5). In our study, this group’s proteins were not identified from the yeast-like fungi, the Agaricomycetes, Kickxellomycotina, and Microsporidia fungi (Appendix A and Figure 5). Interestingly, although this group’s genes identified from other fungal species showed a similar domain structure to those genes from the Pezizomycotina fungi, their sequence lengths were significantly different. However, although the epigenetic activator *ASH1* of *Drosophila* has been characterized as a multi-catalytic histone methyltransferase (HMTase) methylating H3K4, H3K9, H4K20, and H3K36 [50,51], the substrates and functions of the *ASH1*-related proteins in fungi remain unknown. 

##### The SET1 Family (Group 4)

This group of genes commonly exists in all the fungal lineages examined in this study (Appendix A and Figure 5), suggesting their importance in fungi. Besides the conserved N-SET and SET domains at the C-terminus, this group’s proteins also contain a SET-associated (SET-assoc) domain and an RNA recognition motif (RRM) domain at their upstream regions (Figure 5). The RRM domain has been considered to affect the level of K4 methylation [27,52]. CAFE results showed that this group of genes conserved existed in the Ascomycota and Basidiomycota fungi with only one gene copy (Figure 9). However, in the Chytridiomycota fungus *Neoallimastix lanate*, the Zoopagomycotina fungus *Syncephalis fuscata*, and the Mucoromycota fungus *Rhizopus delemar*, this group genes may have suffered expansion (Figure 9). It has been confirmed that the *ScSet1* (*Sc-NP_011987*) is responsible for the overall chromatin mono-, di-, and tri-methylation of histone H3-lysine 4 (H3K4) in *S. cerevisiae*. Thus, we predicted that this group’s genes might conduct the function of H3K4me in fungi. 

##### The SU(Var)3-9 Family (Group 5)

*NcDim-5* (*Nc-EAA28243*) of *Neurospora crassa* and *SpClr4* (*Sp-NP_595186*) of fission yeast *S. pombe*, which have been characterized as Su(var)3-9 family genes, are clustered in this group (Appendix A and Table 1). Thus, group 5 is also named the Su(var)3-9 family. Previous studies showed that Su(var)3-9 family proteins have HKMT activity to conduct the methylation of H3K9 depending on their functional regions in pre-SET [14,53]. The pre-SET domain, which is involved in the structural stability of the SET domain, contains nine invariant cysteine residues, coordinating three zinc ions [54]. Thus, the genes within this group may all have the potential functions to perform the methylation of H3K9. This group’s genes contain a pre-SET motif located upstream of the SET domain (Figure 5). Moreover, several genes such as *SpClr4* (*Sp-NP_595186*) from *S. pombe* and *Ao-AOL_s00076g661* in this group also contain an additional chromodomain at their N-terminus. The Su(var)3-9 family genes in this group were found in the early diverging fungal lineages (Appendix A). However, in the yeast-like fungi, it is interesting that this group’s genes were totally lost in the Saccbaromycotina fungi but still presented in the Taphrinomycotina fungi (Appendix A). In Ascomycota, almost all filamentous fungi contain one copy of this group’s genes. CAFE results showed that these group genes have been lost in the Saccbaromycotina fungi, while gene duplication events have occurred in the Chytridiomycota fungus *Neoallimastix lanate*, the Zoopagomycotina fungus *Linderina pennispora*, and the Agaricomycetes fungus *Suillus tomentosus* and *Armillaria gallica* (Figure 9).

##### The SET2 Family (Group 6)

The representative genes of the Group 6 include the methyltransferase *ScSET2* (*Sc-NP_012367*) from *S. cerevisiae*, *SpSET2* (*Sp-NP_594980*) from *S. pombe*, *NcSET2* (*Nc- XP_957740*) from *N. crassa* (Table 1). Thus, this group is also named as SET2 family. This group’s genes are commonly found in all fungal lineages, with only one gene identified from one of the Microsporidia fungus *Mitosporidium daphniae* while most of the other fungal species employed in this study contain only one gene copy (Appendix A, Figure 5, and Figure 8). It is interesting that our CAFE result showed that the gene expansion only occurred in the Mucoromycota lineage while the other fungal lineages conserved keep this group’s genes, suggesting the importance of this group’s genes (Figure 9). The structure motifs of this group’s proteins show similarity to those of the ASH1-like family (Group 3) proteins, which both process the AWS (Associated With SET), and SET domains. However, the two domains in the *ASH1*-like family were located at the C-terminal, while those domains in the SET2 family were located at the N-terminal (Figure 5). In addition, this group’s proteins also carry an SRI motif (Figure 5), which has been identified to mediate RNA polymerase II interaction [55,56]. In addition, several genes in this group contain another WW-peptide motif and Med26 motif (Figure 5). It is known that the SET2 methyltransferases in the yeasts *S. cerevisiae* and *S. pombe* are responsible for mono-, di-, and tri-methylation of H3K36 [55,57]. The orthologue SET2, which methylates H3K36 in *N. crassa,* is required for normal growth and expression of genes in both the asexual and sexual differentiation pathways [23] 

##### The Group 7-11 Family

Group 7 consists of 31 genes from several fungi belonging to Pezizomycotina fungi except for the three Orbiliomycetes fungal species, suggesting this group’s genes are Pezizomycotina-specific (Appendix A). SET domain, which is located at the N-terminal is the only recognizable conserved structural motif among this group’s genes. It is interesting that the gene derived from the fungus *Paraphoma chrysanthemicola* contains a DEAD and a helicase_C domain at its C-terminal (Figure 5). Group 8 consists of 35 genes from several fungi belonging to Chytridiomycetes, Mucoromycota, and Zoopagomycota fungi (Appendix A). 

Group 9 genes were identified from most of the fungal lineages except for the yeast-like and the Microsporidia fungi. For Group 10, we obtained 85 genes from several fungal species. We only identified Group 11 genes from the Chytridiomycota and Zoopagomycotina fungi. For these four group genes, the SET domain is the only recognizable conserved structural motif among them. Up to now, the functions of these two groups’ genes have not been identified (Figure 5).

##### The SU(Var)4-20 Family (Group 12)

Group 12 genes were represented in most of the fungal lineages except for the Zoopagomycotina fungi. Intriguingly, we did not find the group 12 genes in most of the Saccharomycotina fungi except for *Yarrowia lipolytica*, which contains one gene in this group (Appendix A, Figure 5 and Figure 8). Thus, our CAFE results showed that this group gene was constrained in the Saccharomycotina fungi and Zoopagomycotina fungi, but this group’s genes have expanded in the *Paraphoma chrysanthemicola* because two genes were identified in this fungus (Figure 9). Moreover, in the Mucoromycota fungus *R. delemar* third group, 12 genes were identified (Figure 9). Thus, gene expansion has occurred in this fungus. It is interesting that the putative SET9 homologs (*SpSet9*, *Sp-NP_588078*) identified from the fission yeast *S. pombe* are present in this group (Table 1). This group’s genes only carry only one SET domain located at the N-terminal (Figure 5). It’s known that SpSet9 carries out mono-, di-, and trimethylation of H4K20 and is involved in the maintenance of heterochromatin in metazoans [56,58].

#### 3.4.2. Cluster 2

The genes grouped into Cluster 2 can be divided into six groups (Group 9–13) (Figure 6). Groups 13 and 14 commonly existed in most of the fungal lineages except for Zoopagoycotina and Microsporidia in Group 13 and the Chytridiomycota, Zoopagomycota fungi, and Microsporidia in Group 14 (Figure 8 and Appendix A). For Group 13, the SET domain is the only domain identified while in Group 14, another RubiscoMS domain has been identified downstream of the SET domain. Within Group 13, *SpSet11*(*Sp-NP_588349*) and *Sc-NP_010484* were identified (Table 1). *SpSet13* (*Sp-NP_594072*) and the other two genes (*Nc-XP_958526* and *Sc-NP_010543*) were clustered into Group 14 (Table 1). For Group 15, 22 genes were identified from several Pezizomycotina and Taphrinomycotina fungi. In this group, *N. crassa* (*Nc-XP_960703*) and *SpSet8* (*Sp-NP_593610*) were included in this group (Appendix A and Table 1). As an Ascomycota-specific group, this group’s genes may have emerged after the separation of Ascomycota and Basidiomycota fungi (Figure 8 and Appendix A). Group 16 also mainly consisted of the genes from several Pezizomycotina fungi but also include several genes from Chytridiomycota and Mucoromycota fungi (Figure 8 and Appendix A). Intriguingly, the genes from Pezizomycotina fungi contain a partial SET domain, a RubiscoMS, and a MYND-finger (myeloid, nervy, and DEAF-1 factor), whereas the genes in this group from the Chytridiomycota and Mucoromycota fungi only contain a SET domain (Figure 6), suggesting that the Pezizomycotina fungi have obtained another function during their evolution. MYND motif is a Zn-binding domain involved in H3K36- and H3K4-specific methylation [59,60]. However, although *NcSet9* (*Nc-XP_963594*) from *N. crassa* is also included in this group, nothing is known about its activity or function yet (Table 1). For the genes in Group 17, two genes (*Sc-NP-011824* and *Sc-NP-015116*) from *S. cerevisiae* and the gene *SpSet10* (*Sp-NP-595446*) in *S. pombe* were identified (Table 1). It is interesting that no Group 18 genes were identified from the Pezizomycotina fungi, suggesting that this group’s genes have been lost by the Pezizomycotina fungi.

#### 3.4.3. Cluster 3

In our study, Cluster 3 was divided into nine groups (Group 19–27) (Figure 7). *SpSet11*(*Sp-NP_588361*) from *S. pome* is clustered into group 19 (Table 1). Group 22 genes are only identified from the Eurotiomycetes fungi, suggesting this group’s genes were Eurotiomycetes-specific. Moreover, none of the Group 24 genes were identified from the Pezizomycotina fungi, while no Group 25 genes were identified from the Ascomycota fungi (Figure 8 and Appendix A), indicating that these genes have been lost by the Pezizomycotina and Ascomycota fungi, respectively. Domain structure analysis showed that those genes from Group 25 contain a domain named MAS20 upstream of the SET domain (Figure 7). The genes belonging to Group 21 also carry the MYND domain (Figure 7), suggesting their potential H3K36- and/or H3K4- methylation abilities, though their specific functions are still unknown. The *ScSet5 (Sc-P38890*) from *S. cerevisiae* clustered into Group 21 and it has been identified as an H4K5, K8, and K12 methyltransferase that functions with Set1 to promote repression at telomeres [61]. However, the exact functions of genes belonging to other groups remain unstudied. The gene *ScSET6* (*Sc-Np_015160*) from *S. cerevisiae* belonged to Group 26. Group 21 is the biggest SET-domain-containing gene group in fungi. It contains 541 genes, and the gene numbers range from 0-16 within individual species (Appendix A). Although so many genes are grouped into this Group, their functions are still unknown yet.

### 3.5. Different Families/Groups Have Distinct Evolutionary Histories

At present, the functions of most SET genes identified here have not been characterized, but undoubtedly not all the SET domain proteins are histone methyltransferases. At present, only five clades genes have known histone targets, suggesting that H3K4, H3K9, H3K27, and H3K36 methylation is likely an ancient ability. However, the gene that mediates H3K27 (Group 2) was found in Pezizomycotina fungi but was totally lost in the yeast-like fungi (Appendix A and Figure 8). It is interesting that the genes that mediate H3K9 (Group 5) were also not observed in the Saccharomycotina fungi, and not all the Basidiomycota fungi contain the genes (Appendix A and Figure 8). To study the evolutionary history of the SET-domain genes in fungi, we built phylogenetic trees at the species and gene levels. Using phylogenomic and family sizes to estimate the gene family evolution, the numbers present on each node correspond to the expansion and contraction numbers of orthologous genes predicted by CAFE. As seen in Appendix A, the CAFE analysis showed that there were various expansions and contractions among the fungal lineages (Appendix A). With the further characterization of the five SET domain groups that conduct the function of histone modification in fungi, we found that different gene duplication and loss events have occurred in the five gene groups along with the main evolutionary lineages of fungi, leading to the diversification of their functions in different fungi (Figure 4 and Figure 8). However, for each group’s proteins, too many expansions and contractions have not occurred, which means the fungal species retained a similar gene number of each SET-domain gene, suggesting their conservation and functional importance for fungi. 

Our analysis showed that some SET-domain-containing gene groups are lineage-specific groups because this group’s genes are only present in several specific lineages (Figure 4 and Figure 8). For instance, the genes in Group 5 were all lost in Saccharomycotina fungi and the genes belonging to Chytridiomycota were not identified in Group 1 (Figure 8 and Appendix A). Interestingly, some SET-domain genes were lost in a specific lineage while emerging in a few species of other lineages. For instance, the genes from Groups 7, 15, and 22 are only present in Pezizomycotina fungi, implying that these SET-domain genes have emerged during the evolution of Pezizomycotina lineages (Appendix A and Figure 8). The genes belonging to Groups 8 and 11 were only present in the Zoopagomycota, Chytridiomycota, and other early diverged fungal lineages but disappeared in the Ascomycota and Basidiomycota fungi (Appendix A and Figure 8), suggesting that they were lost by Ascomycota and Basidiomycota fungi. The functions of those genes only presented in specific groups are still unknown. However, they may be expected to modify histones or have very different biological functions during the evolution of fungal species. Thus, studying their functions will be an exciting task. However, although some SET-domain genes within Cluster 2 and Cluster 3 have identified functions, such as function as methylases of ribosomal proteins or elongation factor eEF1, the functions of most are unknown. This is especially the case with some genes that emerged recently and are only present in some lineages, or some genes that are only retained in a few ancestral fungal species.

In general, genes clustered into one clade might have similar functions, whereas novel functions might have formed among the genes clustered into separate clades. The genes within the same group showed highly conserved structured organizations, suggesting they have similar activities (Figure 5, Figure 6 and Figure 7). Thus, although their potential activities have not yet been functionally identified through molecular experiments, we can propose their chemical activities according to the reported activity of homologous genes in the same group. For example, NcDim-5 and SpClr4, which both conduct the methylation of H3K9 were clustered in Group 5 (Figure 5). Thus, we could speculate that the genes that are within this Group but distributed in other fungal species may all mediate the methylation of H3K9. However, more molecular experiments are still required to confirm their specific functions in the future. Interestingly, although the genes within a group have similar sequence and structure organizations, their functions may differ. For example, ScSet3C in *S. cerevisiae* functions as a repressor of meiosis-specific genes [47], mediated by the recruitment of Set3C and ScSet2 by long noncoding RNAs [62,63], the homologs of ScSet3 from *S. cerevisiae* was catalytically inactive due to the conserved NHS tripeptide has been changed to RRS [30], and the Set3C protein from *C. albicans* has been confirmed to play a role in biofilm production and drug resistance [48].

Genes within the same species but belonging to different groups may have obtained different functions. For example, in the model filamentous fungus *N. crassa*, 16 SET-domain genes were identified in our study, and several of them are known to have different functions (Table 1). Previous studies have confirmed that NcDim-5 is responsible for all H3K9me2/3 [46,64], and Ash1-like NcSET-3 may methylate H3K36 [4]. However, the other SET-domain genes in *N. crassa* were neither functionally identified nor activity confirmed.

Furthermore, the genes belonging to different SET groups may have similar functions. For example, the histone methyltransferases Set5 and Set1 share overlapping functions of gene silencing and telomere maintenance in the different fungal SET Groups. In conclusion, although not all the functions of the SET-domain-containing genes have been functionally characterized, those genes may play different roles in increasing adaptability or helping fungi survive in various ecological habitats. 

**Table 1 jof-08-01159-t001:** Distribution of the SET Domain Proteins in *N. crassa*, *S. cerevisiae* and *S. pombe*.

Cluster	Groups	Previous Names	Presence of Methylation	*Neurospora crassa*	*Saccharomyces cerevisiae*	*Schizosaccharomyces pombe*
Cluster 1	Group 1	SET3C family		XP_957466 (SET4)	NP_012430(SET4)/NP_012954 (SET3)	NP_594837 (SET3)
	Group 2	Ez family	H3K27me	XP_965043 (SET7)		
	Group 3	Ash1 family		XP_964116 (SET3)		
	Group 4	SET1 family	H3K4me	XP_961572 (SET1)	NP_011987 (SET1)	NP_587812 (SET1)
	Group 5	SUV39 family	H3K9me	EAA28243 (DIM-5)		NP_595186 (Clr4)
	Group 6	SET2 family	H3K36me	XP_957740 (SET2)	NP_012367 (SET2)	NP_594980 (SET2)
	Group 7					
	Group 8					
	Group 9					
	Group 10					
	Group 11					
	Group 12	Suv4-20 family	H4K20me	EAA33797 (SET10)		NP_588078 (SET9)
Cluster 2	Group 13				NP_010484	NP_588349 (SET11)
	Group 14			XP_958526	NP_010543	NP_594072 (SET13)
	Group 15			XP_960703		NP_593610 (SET8)
	Group 16			XP_963594 (SET9)		
	Group 17				NP_011824/NP_015116	NP_595446 (SET10)
	Group 18				NP_009586	
Cluster 3	Group 19					NP_588361 (SET7)
	Group 20			EAA35550/NCU00870		
	Group 21			XP_957968(SET11)/XP_963161/NCU_00296		NP_588413 (SET5)
	Group 22					
	Group 23					
	Group 24					
	Group 25					
	Group 26				NP_015160(SET6)	
	Group 27			NCU_002962	P38890(SET5)	NP_596514 (SET6)

## 4. Conclusions

So far, only a tiny number of SET-domain genes in fungi have been characterized, and almost all studies have focused on the nonpathogenic model species. Thus, it would be challenging yet fascinating to study the functions and molecular mechanisms of SET-domain genes in those non-model species, especially the important plant and animal pathogens. Our analysis indicated the SET-domain containing genes in fungi had undergone a considerable number of duplications, diversification, and loss events, resulting in broad functional diversities among various fungal species. The evolutionary dynamics help them obtain novel functions, leading them to help fungi adapt to various environments. We expect our study can provide a guideline for the functional research on SET-domain genes of fungi in future studies. 

## Figures and Tables

**Figure 1 jof-08-01159-f001:**
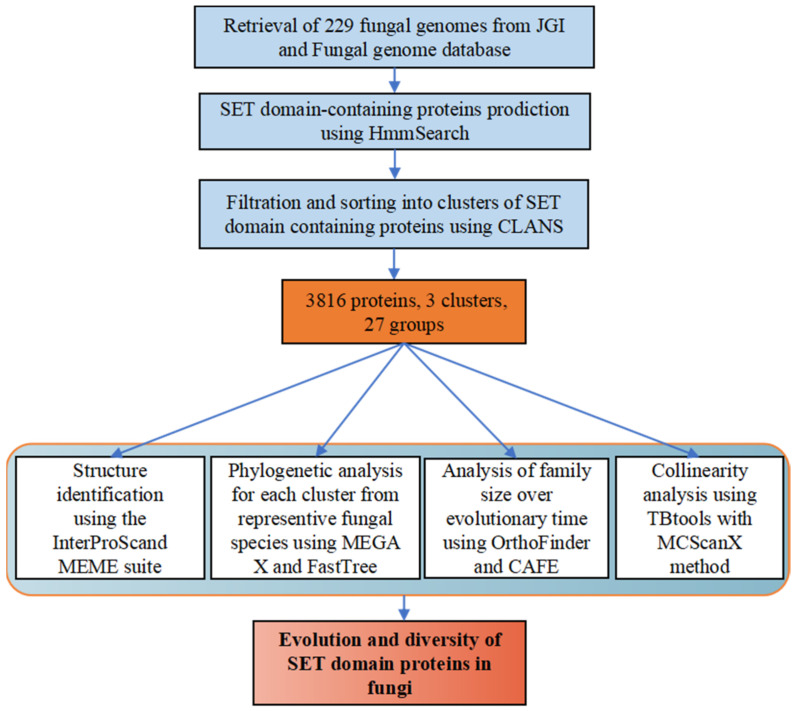
Pipeline schematic of the SET domain proteins in fungi. Predicted protein sets for a given species are fed into the pipeline. In the first step, the protein sets are searched with the SET-domain (PF00856) HMM profile. Next, candidate SET-domain genes are filtered and sorted into families by CLANS program. Then those identified 3816 SET-domain genes were subsequently analyzed with structure analyses, phylogenetic analysis, gene family size analysis, and collinearity analysis.

**Figure 2 jof-08-01159-f002:**
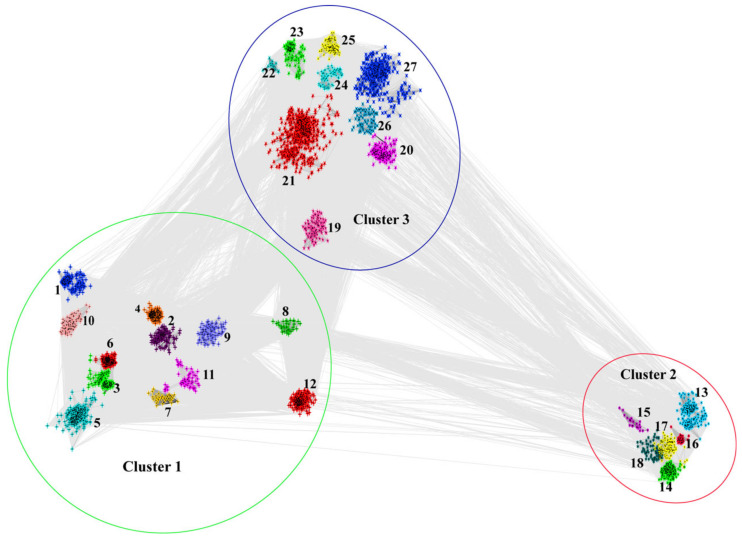
CLANS clustering of 3816 SET domain sequences obtained from 229 whole-genome sequenced fungal species. Three Clusters and 27 groups among the 3816 SET domain sequences were identified. Those 27 groups within these three Clusters were marked with different colors and shapes.

**Figure 3 jof-08-01159-f003:**
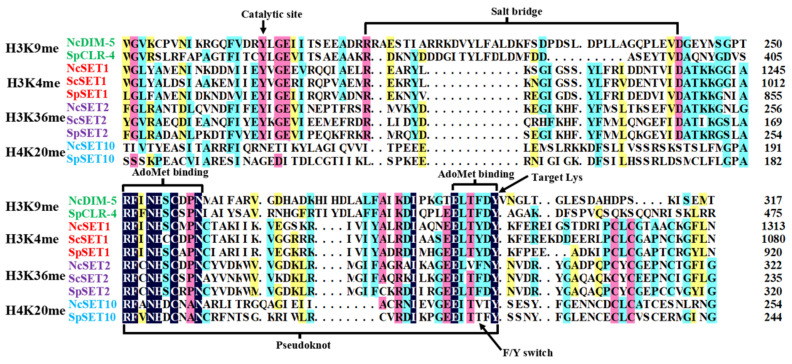
Alignment of several representative SET-domain-containing histone lysine methyltransferases (HKMT) grouped according to their histone-lysine specificity from N. crassa (Nc), S. cerevisiae (Sc), and S. pombe (Sp). Those HKMTs all belong to Cluster 1 in Figure 2 with NcDIM-5 and SpCLR-4 belonging to Group 5, NcSET1, ScSET1, and SpSET1 belonging to Group 4, NcSET2, ScSET2, and SpSET2 belonging to Group 6, and NcSET10 and SpSET1 belonging to Group 12. The residues of the AdoMet, target lysine, catalytic site, the structural pseudoknot, an intra-molecular interacting salt bridge, and a F/Y switch controlling whether the product is a mono-, di- or tri-methylated histone are indicated.

**Figure 4 jof-08-01159-f004:**
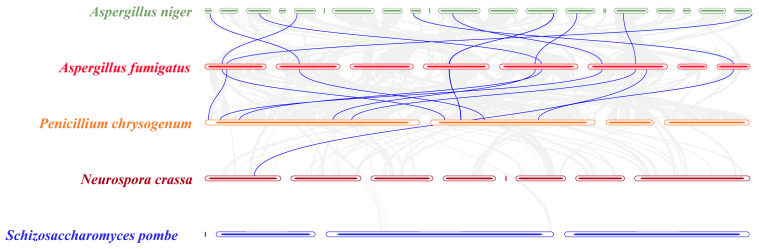
Collinearity analysis of the SET-domain genes from five species. A total of 13 SET-domain genes derived from A. fumigatus (AFUA_1G03000, AFUA_1G11090, AFUA_2G08510, AFUA_2G08775, AFUA_2G10080, AFUA_3G06400, AFUA_3G06480, AFUA_4G09180, AFUA_5G06000, AFUA_5G12710, AFUA_6G04520, AFUA_6G06335, and AFUA_7G04410) to show the collinearity relationships with the orthologous from other four species. These 13 SET-domain genes were distributed into all the three clusters from Figure 2. The gray lines indicate gene blocks in A. fumigatus that are orthologous to the other genomes. The blue lines delineate the syntenic SET-domain gene pairs.

**Figure 5 jof-08-01159-f005:**
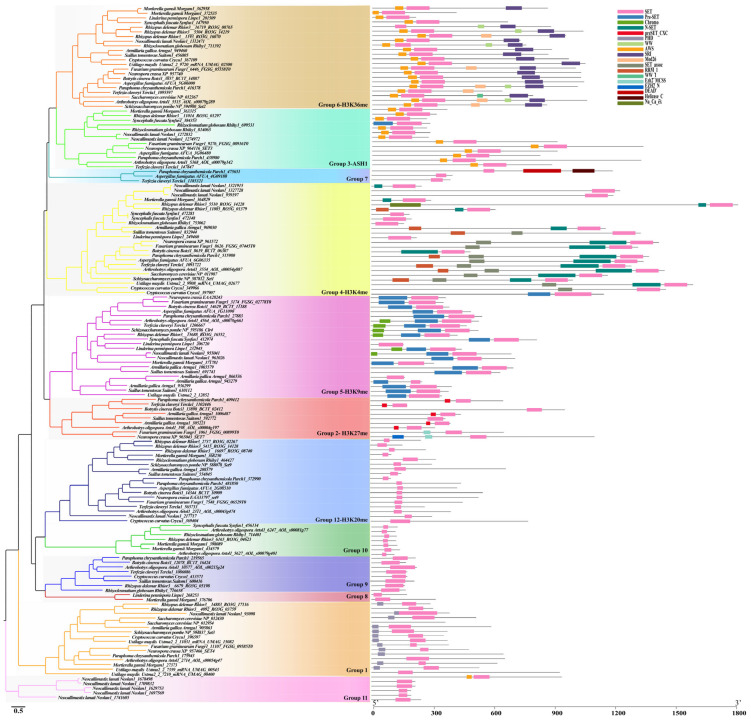
Phylogenetic analysis and domain organization of the Cluster 1 SET domain sequences from 20 representative 20 fungal species. The protein domains were labeled with different colors based on the domain type. Domain organization of SET-domain-containing proteins was detected by the InterProScan 5.0 (http://www.ebi.ac.uk/interpro/) and MEME suite (https://meme-suite.org/meme/) with default parameter settings. Phylogenetic analyses were performed using NJ with MEGA X and ML analysis with FastTree.

**Figure 6 jof-08-01159-f006:**
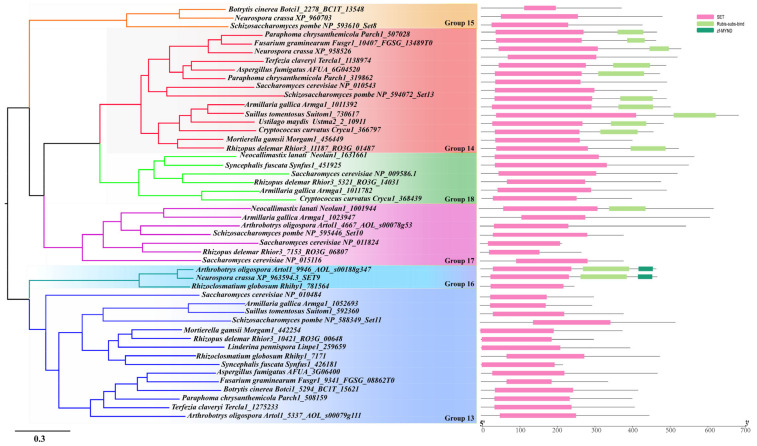
Phylogenetic analysis and domain organization of the Cluster 2 SET domain sequences from 20 representative 20 fungal species. The protein domains were labeled with different colors based on the domain type. Domain organization of SET-domain-containing proteins was detected by InterProScan 5.0 (http://www.ebi.ac.uk/interpro/) and MEME suite (https://meme-suite.org/meme/) with default parameter settings. Phylogenetic analyses were performed using NJ with MEGA X and ML analysis with FastTree.

**Figure 7 jof-08-01159-f007:**
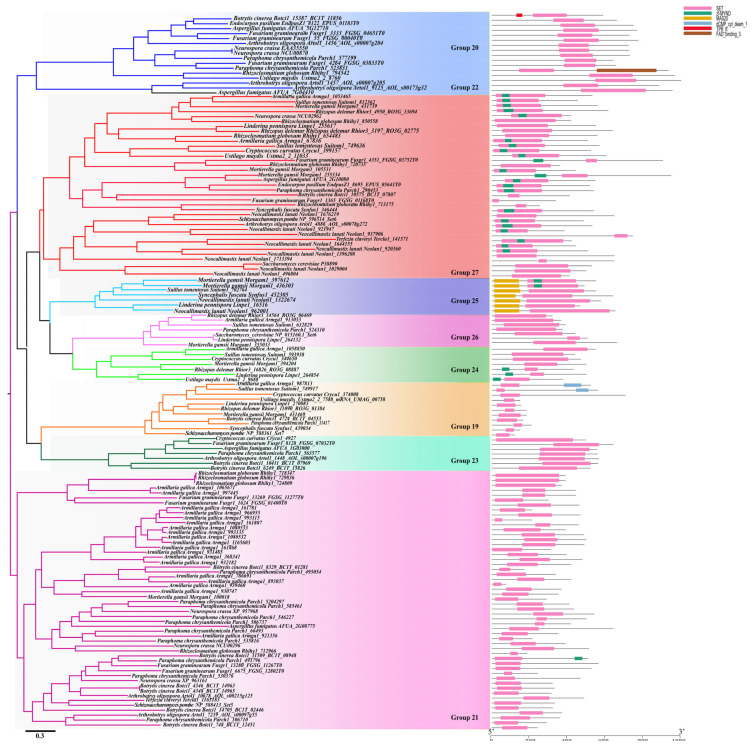
Phylogenetic analysis and domain organization of the Cluster 3 SET domain sequences from 20 representative 20 fungal species. The protein domains were labeled with different colors based on the domain type. Domain organization of SET domain containing proteins were detected by the InterProScan 5.0 (http://www.ebi.ac.uk/interpro/) and MEME suite (https://meme-suite.org/meme/) with default parameter settings. Phylogenetic analyses were performed using NJ with MEGA X and ML analysis with FastTree.

**Figure 8 jof-08-01159-f008:**
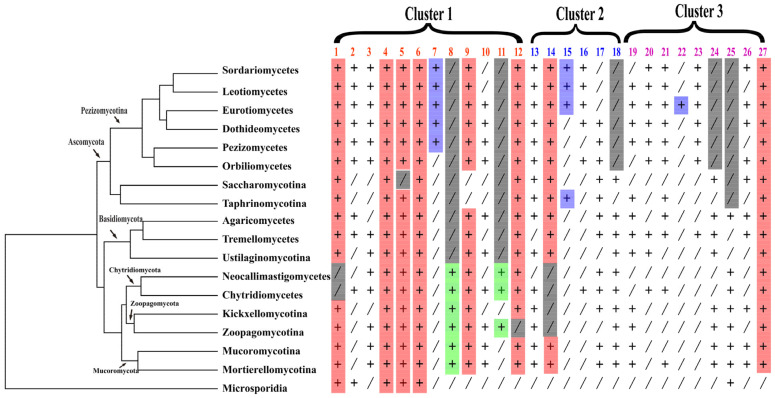
Diversification of SET-domain genes in fungi. Distribution pattern of SET genes in different fungal taxonomic groups. Those 3816 SET-domain genes from 229 fungal species can mainly be classified into three distinct clusters (Cluster 1, 2, and 3, respectively), and twelve, six, and nine groups can be observed in each of the three clusters. One representative is elected for each order. “Plus” indicates the presence of SET genes, and “Slash” indicates the absence of SET genes.

**Figure 9 jof-08-01159-f009:**
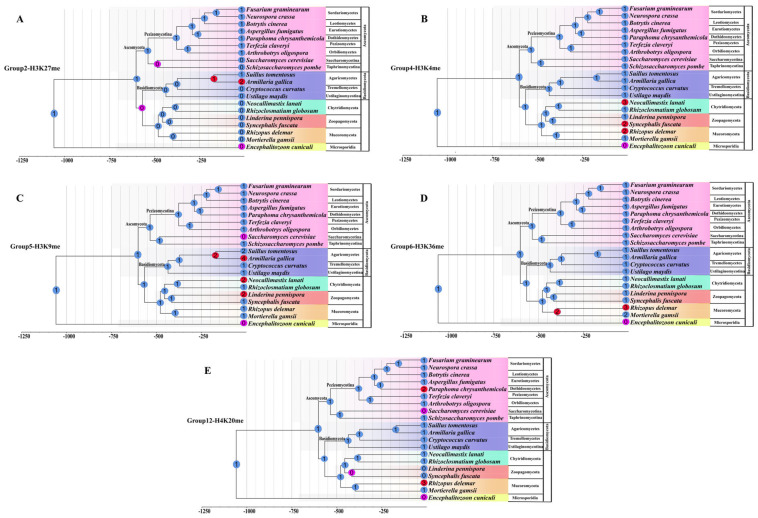
The evolution of five groups SET-domain genes analyzed with CAFE. Each tip of the tree represents a species. The numbers present on each node correspond to the number of SET-domain genes predicted by CAFE. Blue = No change; Red = Expansion; Purple = Contraction.

## Data Availability

Not applicable.

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
