# Peer review of "Phylogenomic and Evolutionary Analyses Reveal Diversifications of SET-Domain Proteins in Fungi"

_jof, 2022, doi:10.3390/jof8111159_

Round 1

Reviewer 1 Report

Dear Guoqing Ding and colleagues,

I read your analysis of the SET domain in different fungi with great interest.

Here are my comments:

Please keep in mind that the readership may not only include experts in the field. The more explanation is included, the wider the readership.

Page 3: The readership would benefit from a brief summary of the key features of a SET domain (e.g. an image showing a SET domain with the key residues highlighted)

Page 3: please explain why it is important to remove the separate points in the CLANS program. Say here what the programm does.

Page 4: That some fugal species have no SET domain is interesting. Please comment on how lysine methyltransferase activity could work in the absence of a SET domain. Is there a correlation between fungal biology and the number of SET domains?

Page 5: Please explain why A.fumigatus is here used as an example. Please explain TBtools. Please explain collinearity relationship.

Page 11: Please explain "group 12". In the legend of Figure 7, please explain what the clusters 1-3 are.

Page 14: Please explain CAFE analysis.

Author Response

Dear Reviewer,

Thank you for providing us an opportunity to revise our manuscript (Manuscript ID:jof-1956913). We have carefully revised the manuscript according to your comments. We hope the revised manuscript can meet your requirements. Please see our "Response to Reviewer" as follows:

Sincerely Yours,

Juan Li

Reviewer #1

Page 3: The readership would benefit from a brief summary of the key features of a SET domain (e.g. an image showing a SET domain with the key residues highlighted)

Reply: Thank you for your comments. In the revised ms, we added more description on the key features of SET domains. We also provided a new image that show the key residues of SET domain in the revised ms. To show the key features of a SET domain, we did an alignment of several representative histone lysine methyltransferases (HKMT) grouped according to their histone-lysine specificity. The alignment showed that the SET domain sequences contained two most-conserved sequence motifs (RFINHXCXPN and ELXFDY; see Figure 3 in the revised ms) binding to S-adenosylmethionine (AdoMet)and the target lysine of the SET domain. Moreover, the residues of catalytic site, an intra-molecular interacting salt bridge, and a F/Y switch controlling whether the product is a mono-, di- or tri-methylated histone were also identified. Previous studies supposed that the two conserved motifs binding to AdoMet were brought together by the “pesudoknot” fold to form an active site in a location immediately next to the pocket where the methyl donor binds and to the peptide-binding cleft. Most notably, the substrate-binding clefts are connected through a deep channel that runs through the core of the SET domain and permits transfer of the methyl group from AdoMet to the -amino group of the lysine substrate.

Page 3: please explain why it is important to remove the separate points in the CLANS program. Say here what the programm does.

Reply: Thank you for your comments. CLANS is a Java-based software program which uses the Fruchterman-Reingold graph layout algorithm to visualize pairwise sequence similarities in either two-dimensional or three-dimensional space. The program performs all-against-all BLAST searches and calculates pairwise attraction values based on the HSP P-values. The graph showing all pairwise interactions can be rotated, translated and zoomed to better view sequence relationships. Homologous relationships are typically established through sequence similarity searches, multiple alignments and phylogenetic reconstruction. Unlike phylogenetic reconstruction it becomes more accurate with an increasing number of sequences, as the larger number of pairwise relationships average out the spurious matches that are the crux of simpler pairwise similarity-based analyses. In conclusion, CLANS was developed to analyze relationships in large sequence datasets. In cases where a sequences belong to one family should have similar or highly sequence similarities, whereas a sequence showed low sequence similarities with other sequences would be set as separated point in the CLANS program. So those sequences showed low sequence similarities with other sequences would be formed as the separated points in CLANS program. Thus, to understand the relationships among different SET domain-containing families, we removed the separate points in the CLANS program.

Page 4: That some fugal species have no SET domain is interesting. Please comment on how lysine methyltransferase activity could work in the absence of a SET domain. Is there a correlation between fungal biology and the number of SET domains?

Reply: Thank you for your comments. In the revised ms, we did some discussions on how lysine methyltransferase activity could work in the absence of a SET domain. The explanation on why those ancestral Microsporidia fungi did not contain any SET domain genes may be associated with their genome size. As we can see in Table S1, all the genome sizes of Microsporidia fungal species were less than 10Mb, suggesting that these species have only retained those essential functional genes to maintain their life. Previous studies in Arabidopsis has revealed the importance of DNA methylation in regulating histone methylation [41]. In addition, scientists have found that the yeast Cryptococcus neoformans has lost the methylase gene between 50-150 mya, but the methylation in C. neoformans has persisted because an epigenome has been propagated for >50 million years through a process analogous to Darwinian evolution of the genome [42,43]. Thus, although it is still hard to explain how the lysine methyltransferase activity could work in the fungi without SET domain gene, we supposed that there may some uncovered mechanisms for those fungi to active their lysine methyltransferase. 

  1. 41. Mendes, F.K.; Vanderpool, D.; Fulton, B.; Hahn, M.W. CAFE 5 models variation in evolutionary rates among gene families. Bioinformatics 2020, doi:10.1093/bioinformatics/btaa1022.
  2. 42. Emms, D.M.; Kelly, S. OrthoFinder: phylogenetic orthology inference for comparative genomics. Genome Biol 2019, 20, 238, doi:10.1186/s13059-019-1832-y.
  3. 43. Sanderson, M.J. r8s: inferring absolute rates of molecular evolution and divergence times in the absence of a molecular clock. Bioinformatics 2003, 19, 301-302, doi:10.1093/bioinformatics/19.2.301.

Page 5: Please explain why A. fumigatus is here used as an example. Please explain TBtools. Please explain collinearity relationship.

Reply: Thank you for your comments. In the revised ms, we used A. fumigatus genome annotation information to visualize the chromosomal distribution of the SET-domain containing genes. Moreover, another reviewer suggested we use the chromosomal level assembled genomes to do the collinearity analysis. Thus, we also explored the collinearity relationships of the SET-domain genes among five species, including A. fumigatus, Aspergillus niger, Penicillium chrysogenum, Neurospora crassa, and Schizosaccharomyces pombe. The reason that we used those fungi as example is that all these five fungal species were model species and have been extensively studied previously. Moreover, the genome of these five species were well assembled, which can provide more confirmed information for understanding the collinearity relationships of SET domains. Also, we also explained TBtools and collinearity relationships in the revised ms.

Page 11: Please explain "group 12". In the legend of Figure 7, please explain what the clusters 1-3 are.

Reply: Thank you for your comments. We explained what the clusters 1-3 are in the revised Figure 7. Moreover, Group 12 also were explained in the revised ms.

Page 14: Please explain CAFE analysis.

Reply: Thank you for your comments.We explained CAFE analysis in the revised ms.

Reviewer 2 Report

The paper submitted by Ding and co-authors systemically identified and characterized SET-domain containing proteins in 229 fungal genomes. They have identified 3816 SET-domain proteins in total, which could be classified into 3 main clusters and 27 groups. Together, this study has illustrated the evolution and diversification of SET-domain containing proteins in fungi, and will be of great interests and valuable resource to researchers working on epigenetics. The major flaw of this study is that the completeness of genome assemblies used in this study is not fully evaluated, which may lead to inaccuracy of the analysis.

L53-L66, the regulatory roles of the SET proteins should be given if available. This information will be very helpful for readers that not work on fungi. For example, some SET proteins are involved in suppression/activation of TE or gene expression, whether their roles are conserved in fungi should be given.

L133, the authors should confirmed all genome sequences downloaded from JGI can be used for publication, since some of the publicly available genomes from JGI need specific authorization for publication.

Figure 1, The completeness of the genome assemblies will be very important for the annotation of the SET proteins. I would suggest the authors to exclude these low-quality assemblies in the first step.

It's very interesting that some Microsporidia genomes, such as Hepatospora eriocheir, only contains 1 or no SET protein. Is it because they are too far away phylogenetically from other fungi?

Table S1, the meaning of number 1 to 27 in the first row should be given and the potential modifications mediated by these groups could also be included. Assembly quality information, such as genome size, contig number, N50 etc, should be included in this table. The genome size and number of SET proteins identified could be another evidence showing expansion or contraction of a gene family.

Figure 3, Collinearity analysis is essential to infer duplication and structural variation events in a chromosomal view. However, the assemblies of A. oligospora, D. stenobrocha, and M. haptotylum are highly fragmented. I would suggest the authors use chromosomal level assemblies, like N. crassa and genome of Figure S2, for this analysis. I also noticed that there are no grey lines between A. oligospora and A. cepistipes, which is very unlikely to happen in this analysis.

Minor point:

Figure 7, Grey lines could be removed at the phylogenetic tree side to make the figure more clear. 

Author Response

Dear Reviewer,

Thank you for providing us an opportunity to revise our manuscript (Manuscript ID:jof-1956913). We have carefully revised the manuscript according to your comments. We hope the revised manuscript can meet your requirements. Please see our "Response to Reviewer" as follows.

Sincerely Yours,

Juan Li

Reviewer #2

The paper submitted by Ding and co-authors systemically identified and characterized SET-domain containing proteins in 229 fungal genomes. They have identified 3816 SET-domain proteins in total, which could be classified into 3 main clusters and 27 groups. Together, this study has illustrated the evolution and diversification of SET-domain containing proteins in fungi, and will be of great interests and valuable resource to researchers working on epigenetics. The major flaw of this study is that the completeness of genome assemblies used in this study is not fully evaluated, which may lead to inaccuracy of the analysis.

L53-L66, the regulatory roles of the SET proteins should be given if available. This information will be very helpful for readers that not work on fungi. For example, some SET proteins are involved in suppression/activation of TE or gene expression, whether their roles are conserved in fungi should be given.

Reply: Thank you for your comments. In the revised ms, the regulatory roles of the SET proteins in fungi have added. In fungi, the posttranslational modification of histone N-terminal tails is overall conserved. In general, the histone modification landscape usually generates transcription-ally active or transcriptionally repressive. The cross talk between various modification machineries formed interconnected positive and negative feedback loops that control gene regulation [13]. It has been frequently asserted that H3K9me is associated with the silent regions of both euchromatin and heterochromatin, and responsible for transcriptional re-pression [14-16], while H3K4me is related exclusively to actively transcribed genes[17]. In addition, H3K36me which is catalyzed by SET2 is also associated with gene activity [18]. Those histone lysine modifications have been extensively investigated to play vital roles in fungal development and various biological processes.In fungi, the posttranslational modification of histone N-terminal tails is overall conserved. In general, the histone modification landscape usually generates transcription-ally active or transcriptionally repressive. The cross talk between various modification machineries formed interconnected positive and negative feedback loops that control gene regulation [13]. It has been frequently asserted that H3K9me is associated with the silent regions of both euchromatin and heterochromatin, and responsible for transcriptional re-pression [14-16], while H3K4me is related exclusively to actively transcribed genes[17]. In addition, H3K36me which is catalyzed by SET2 is also associated with gene activity [18]. Those histone lysine modifications have been extensively investigated to play vital roles in fungal development and various biological processes.

  1. Zhang, T.; Cooper, S.; Brockdorff, N. The interplay of histone modifications - writers that read. EMBO Rep 2015, 16, 1467-1481, doi:10.15252/embr.201540945.
  2. Rea, S.; Eisenhaber, F.; O'Carroll, D.; Strahl, B.D.; Sun, Z.W.; Schmid, M.; Opravil, S.; Mechtler, K.; Ponting, C.P.; Allis, C.D.; et al. Regulation of chromatin structure by site-specific histone H3 methyltransferases. Nature 2000, 406, 593-599, doi:10.1038/35020506.
  3. Nakayama, J.; Rice, J.C.; Strahl, B.D.; Allis, C.D.; Grewal, S.I. Role of histone H3 lysine 9 methylation in epigenetic control of heterochromatin assembly. Science 2001, 292, 110-113, doi:10.1126/science.1060118.
  4. Grewal, S.I.; Moazed, D. Heterochromatin and epigenetic control of gene expression. Science 2003, 301, 798-802, doi:10.1126/science.1086887.
  5. Sims, R.J., 3rd; Reinberg, D. Histone H3 Lys 4 methylation: caught in a bind? Genes Dev 2006, 20, 2779-2786, doi:10.1101/gad.1468206.
  6. Bilokapic, S.; Halic, M. Nucleosome and ubiquitin position Set2 to methylate H3K36. Nat Commun 2019, 10, 3795, doi:10.1038/s41467-019-11726-4.

L133, the authors should confirmed all genome sequences downloaded from JGI can be used for publication, since some of the publicly available genomes from JGI need specific authorization for publication.

Reply: Thank you for your comments. We have confirmed all the genome sequences we downloaded from JGI can be used for publication. As you see in Table S1, most of the genomes we employed have been published, and we also double checked those genome data without paper publication to make sure that all the data can be used in our analysis.

Figure 1, The completeness of the genome assemblies will be very important for the annotation of the SET proteins. I would suggest the authors to exclude these low-quality assemblies in the first step.

Reply: Thank you for your comments. We agreed with your comment that the the genome assemblies will be very important for the annotation of the SET proteins. In the revised ms, we collected the assembly quality information, such as genome size, contig number, genome coverage, N50 etc. for each genome. As you can see in the Table S1, the fungal genome we employed in this manuscript were highly assembled. Moreover, in our analysis, we employed more than one fungal species belong to same taxonomy classification to investigated the SET domain proteins and the distribution of SET domain showed highly similarities in the fungal species from same taxonomy classification. In addition, our subsequently analysis were mainly based on the 20 representative fungal species (Red marked in the Table S1). These 20 representative fungal species were also highly assemblies, some of them even genome completed or assembled on chromosomal level. Thus, we believed our results were credible.

It's very interesting that some Microsporidia genomes, such as Hepatospora eriocheir, only contains 1 or no SET protein. Is it because they are too far away phylogenetically from other fungi?

Reply: Thank you for your comments. We identified the the putative SET-domain containing sequences from fungi by using the HMM profile SET domain.Thus, we believed that there is no bias for our searching results. The reason that some Microsporidia genomes only contains 1 or no SET protein in our results may be explained that those ancestral fungi have lost the SET domain genes during their evolution. In the revised ms, we did some discussions on how lysine methyltransferase activity could work in the absence of a SET domain. The explanation on why those ancestral Microsporidia fungi did not contain any SET domain genes may be associated with their genome size. As we can see in Table S1, all the genome sizes of Microsporidia fungal species were less than 10Mb, suggesting that these species have only retained those essential functional genes to maintain their life. Previous studies in Arabidopsis has revealed the importance of DNA methylation in regulating histone methylation [41]. In addition, scientists have found that the yeast Cryptococcus neoformans has lost the methylase gene between 50-150 mya, but the methylation in C. neoformans has persisted because an epigenome has been propagated for >50 million years through a process analogous to Darwinian evolution of the genome [42,43]. Thus, although it is still hard to explain how the lysine methyltransferase activity could work in the fungi without SET domain gene, we supposed that there may some uncovered mechanisms for those fungi to active their lysine methyltransferase.

  1. Zhao, L.; Zhou, Q.; He, L.; Deng, L.; Lozano-Duran, R.; Li, G.; Zhu, J.K. DNAmethylation underpins the epigenomic landscape regulating genome transcription inArabidopsis. Genome Biol 2022, 23, 197, doi:10.1186/s13059-022-02768-x.
  2. Catania, S.; Dumesic, P.A.; Pimentel, H.; Nasif, A.; Stoddard, C.I.; Burke, J.E.; Diedrich, J.K.; Cook, S.; Shea, T.; Geinger, E.; et al. Evolutionary Persistence of DNA Methylation for Millions of Years after Ancient Loss of a De Novo Methyltransferase. Cell 2020, 180, 263-277.e220, doi:10.1016/j.cell.2019.12.012.
  3. Du, J.; Johnson, L.M.; Jacobsen, S.E.; Patel, D.J. DNA methylation pathways and their crosstalk with histone methylation. Nat Rev Mol Cell Biol 2015, 16, 519-532, doi:10.1038/nrm4043.

Table S1, the meaning of number 1 to 27 in the first row should be given and the potential modifications mediated by these groups could also be included. Assembly quality information, such as genome size, contig number, N50 etc, should be included in this table. The genome size and number of SET proteins identified could be another evidence showing expansion or contraction of a gene family.

Reply: Thank you for your comments. In the revised ms, we provided the meaning of number 1 to 27 in the revised Table S1. Moreover, the assembly quality information, such as genome size, contig number, genome coverage, N50 etc. also have been added in Table S1. Also, we did some discussions on the potential relationships of  genome size and the number of SET domain proteins in the revised ms.  

Figure 3, Collinearity analysis is essential to infer duplication and structural variation events in a chromosomal view. However, the assemblies of A. oligospora, D. stenobrocha, and M. haptotylum are highly fragmented. I would suggest the authors use chromosomal level assemblies, like N. crassa and genome of Figure S2, for this analysis. I also noticed that there are no grey lines between A. oligospora and A. cepistipes, which is very unlikely to happen in this analysis.

Reply: Thank you for your comments. In the revised ms, we did the collinearity analysis of the SET-domain genes from five species (A. fumigatus, Aspergillus niger, Penicillium chrysogenum, Neurospora crassa, and Schizosaccharomyces pombe). The reason that we used those fungi as example is that all these five fungal species were model species and have been extensively studied previously. Moreover, the genome of these five species were well assembled, which can provide more confirmed information for understanding the collinearity relationships of SET domains.

Minor point:

Figure 7, Grey lines could be removed at the phylogenetic tree side to make the figure more clear.

Reply: Thank you for your comments. We removed the grey lines of Figure 7 in the revised ms.

Round 2

Reviewer 2 Report

The authors have fully addressed most of my comments.

Some minor comments left:

1 L53-61: H3K27me3 is a more well studied histone modification in fungi.

2 L183: reference 41 is not very relavant to the topic discussed. The main idea of the cited paper is about dramatic changes of histone modifications after complete DNA methylation loss.

Author Response

Dear Reviewer, Thank you for providing us an opportunity to revise our manuscript (Manuscript ID:jof-1956913). We have carefully revised the manuscript according to your comments. We hope the revised manuscript can meet your requirements. Sincerely Yours, Juan Li The authors have fully addressed most of my comments. Some minor comments left: 1 L53-61: H3K27me3 is a more well studied histone modification in fungi. Reply: Thank you for your comments. In the revised ms, the regulatory roles of H3K27me3 in fungi have added. Trimethylation of histone H3 lysine 27 (H3K27me3) had been recognized as im-portant for differentiation and development in fungi[19,20]. Aside from its relatively well understood role in transcriptional repression, accumulating evidence suggests that H3K27 methylation has an important role in controlling the balance between maintenance and generation of novelty in fungal genomes [21]. Reference : 19. Jamieson, K.; Rountree, M.R.; Lewis, Z.A.; Stajich, J.E.; Selker, E.U. Regional control of histone H3 lysine 27 methylation in Neurospora. Proc Natl Acad Sci U S A 2013, 110, 6027-6032, doi:10.1073/pnas.1303750110. 20. Connolly, L.R.; Smith, K.M.; Freitag, M. The Fusarium graminearum histone H3 K27 methyltransferase KMT6 regulates development and expression of secondary metabolite gene clusters. PLoS Genet 2013, 9, e1003916, doi:10.1371/journal.pgen.1003916. 21. Ridenour, J.B.; Möller, M.; Freitag, M. Polycomb Repression without Bristles: Facultative Heterochromatin and Genome Stability in Fungi. Genes (Basel) 2020, 11, doi:10.3390/genes11060638. 2 L183: reference 41 is not very relavant to the topic discussed. The main idea of the cited paper is about dramatic changes of histone modifications after complete DNA methylation loss. Reply: Thank you for your comments. We deleted this reference in the revised ms.